# Performance-Guaranteed ODE Solvers with Complexity-Informed Neural Networks

**Feng Zhao**
Department of Electronic Engineering, Tsinghua University
Noah's Ark Lab, Huawei, Beijing, China
zhaof17@mails.tsinghua.edu.cn

**Xiang Chen**
Noah's Ark Lab, Huawei
Beijing, China
xiangchen.ai@huawei.com

**Jun Wang**
University College London, London, United Kingdom
jun.wang@cs.ucl.ac.uk

**Zuoqiang Shi**
Department of Mathematical Sciences, Tsinghua University, Beijing, China 10084
Yanqi Lake Beijing Institute of Mathematical Sciences and Applications
Beijing, China 101408 zqshi@mail.tsinghua.edu.cn

**Shao-Lun Huang** *
DSIT Research Center
Tsinghua-Berkeley Shenzhen Institute, Shenzhen, China 518055
shaolun.huang@sz.tsinghua.edu.cn

## Abstract

Traditionally, we provide technical parameters for ODE solvers, such as the order, the stepsize and the local error threshold. However, there is no guarantee for performance metrics that users care about, such as the time consumption and the global error. In this paper, we provide such a user-oriented guarantee by using neural networks to fit the complex relationship between the technical parameters and performance metrics. The form of the neural network is carefully designed to incorporate the prior knowledge from time complexity analysis of ODE solvers, which has better performance than purely data-driven approaches. We test our strategy on some parametrized ODE problems, and experimental results show that the fitted model can achieve high accuracy, thus providing error guarantee for fixed methods and time guarantee for adaptive stepsize methods.

## 1 Introduction

The choice of stepsize for ODE solvers has great influences on the computational time and error, which are two fundamental performance metrics when solving differential equations. Existing approaches to choose the stepsize usually make the solver satisfy either the time or error constraint by providing some technical parameters. On the one hand, the fixed stepsize method like forward Euler is often used in real-time simulator of complex systems [1], which requires strict time guarantee of the method. On the other hand, the adaptive stepsize method with error control strategy is used in settings when the precision of the solution is required. In both cases, no convenient method has been

---

*corresponding author

35th Conference on Neural Information Processing Systems (NeurIPS 2021), Sydney, Australia.

developed to choose the appropriate hyper-parameters of ODE solvers to satisfy the requirements. In recent years, artificial neural networks are developed for the design of new ODE solvers [2] or the construction of differentiable integrators (NeuralODE)[3] to facilitate the parameter identification of dynamic systems. Based on the pioneering work of NeuralODE, the proxy ODE simulators (hypersolvers) are constructed using neural networks to reduce the time complexity of the inference [4]. To our best knowledge, no efforts have been spared to use neural networks for the purpose of guaranteeing the user-oriented performance. In our work, we will fill this gap by selecting ODE solvers with complexity-informed neural networks, which is a combination of the mathematical model and the data-driven approach. To unify the two control goals, we utilize neural networks to fit the complex relationship between user-oriented metrics (the error or time) and technical parameters for given ODE problems. Using our approach, we can provide strict global error control for fixed stepsize method and approximated time guarantee for adaptive stepsize method, which enlarges the usability for existing ODE solvers.

## 2   Methodology

We consider solving the initial value problem for a given ODE:

$$
\frac{du}{dt} = f(t, u)
$$
$$
u(t_0) = u_0
$$

(1)

Many solvers are available to obtain numerical solutions of $u(t)$ for an interval $[t_0, t_{\max}]$. For explicit Runge-Kutta (RK) methods, $u_n$ is updated with the following scheme to approximate the value $u(t_n)$.

$$
u_{n+1} = u_n + h_n \psi(t_n, u_n, h_n)
$$
$$
t_{n+1} = t_n + h_n
$$

(2)

In the above formula, $\psi(t, u, h)$ is a scheme-specific function for different RK methods. For example, forward Euler method has $\psi(t, u, h) = f(t, u)$. Depending on whether $h_n$ changes or not, the methods can be divided by the fixed stepsize method ($h_n = h$ is pre-specified) and the adaptive stepsize method ($h_n$ is adjusted at each step).

In the above, we have described the numerical routine to provide the values of two performance metrics $Q_1$ and $Q_2$. For these two metrics, our goal is to find their optimal tradeoff, which is controlled by varying the threshold (abbreviated as th). Let $g_{\text{th}}$ be a function which maps a problem (parametrized by $p$) to the most suitable solver (parametrized by $s$, which is specified by a certain integration method and stepsize control strategy). Then for a given th, $g_{\text{th}}$ is determined by

$$
\max_{g_{\text{th}}:p \to s} \quad Q_1(p, g_{\text{th}}(p))
$$
$$
\text{s.t.} \quad Q_2(p, g_{\text{th}}(p)) \geq \text{th}
$$

(3)

The optimal $g_{\text{th}}$ for (3) makes us achieve the best for the first performance metric $Q_1$, while satisfying the threshold requirement for the second performance metric $Q_2$. Within this article, we focus on two performance metrics: accuracy (global error) and efficiency (time complexity).

The function $g_{\text{th}}$ should be pre-trained and well generalized on problems, so that in usage, we don't need to spend extra computational cost.

### 2.1   Global error control for fixed stepsize method

The time complexity for fixed stepsize method is inverse proportional to the adopted stepsize $h$ while the relationship between the error and $h$ is unknown. To provide error control in such a case, we replace abstract variables of (3) in the following way: $Q_1$ specially means $T_1$, $Q_2$ means the global error (abbreviated as err), and th means the error tolerance (abbreviated as tol).

$$
\min_h \quad T(p, \text{RK}, h)
$$
$$
\text{s.t.} \quad \text{err}(p, \text{RK}, h) \leq \text{tol}
$$

(4)

Once we fix the $p$ and RK method in use, the minimal $T$ is reached when the equality constraint is satisfied. We then need to find an inverse mapping from tol to $h$ such that $\text{err}(p, \text{RK}, h) = \text{tol}$.

Since the local truncation error is of the order $O(h^q)$ where $q$ is the order of the RK method, the relationship between $\log h$ and $\log(\text{err})$ is approximately linear. Therefore, we assume the inverse mapping has the following form

$$\log h = k \log(\text{err}) + C(p) \tag{5}$$

For a given problem parametrized by $p$, the intercept function $C(p)$ (represented by a neural network) and the slope $k$ can be learned after obtaining multiple triads of $(p_i, h_i, \text{err}_i)$. In latter experiments, we show that (5) performs better than fitting the functional relationship from $(h, \text{err})$ to $h$ directly by an MLP with the same model complexity. The loss function can be constructed by MSE or the quantile loss. When more strict error control is preferred, the latter loss is adopted for a small quantile value $q$. In inference stage, the learned model is used directly to obtain the step $h$ given the prescribed error level tol for fixed stepsize method.

To interpret the learned model visually, we consider solving a simple ODE problem by classical RK with order 4 [5]. The integration interval is $[0, t_{\max}]$, and the solution is a two-dimensional spiral.

$$\begin{aligned}
\frac{du_1}{dt} &= b \cos t - u_2 + u_2(t_0), u_1(t_0) = x_0 \\
\frac{du_2}{dt} &= b \sin t + u_1 - u_1(t_0), u_2(t_0) = y_0
\end{aligned} \tag{6}$$

For this ODE problem, the parameter $p$ is concatenated by $\{b, x_0, y_0, t_{\max}\}$. To train the model described by (5), we first sample multiple data of $(p, h)$ from bounded intervals and obtain the corresponding global errors in $\ell_\infty$ norm. Using an MLP with one hidden layer to describe $C(p)$, we obtain the model illustrated in Figure 1 by varying one input variable at each time. Since (5) captures the order condition, the fitted curve matches the true relationship exactly as shown in (a). For other inputs, (5) differs not far from the numerical approximation of the true relationship, as shown in (b,c). By using the quantile loss, all the fitted curves in this example problem lie below the corresponding true one, thus the strict error control is achieved in such a case.

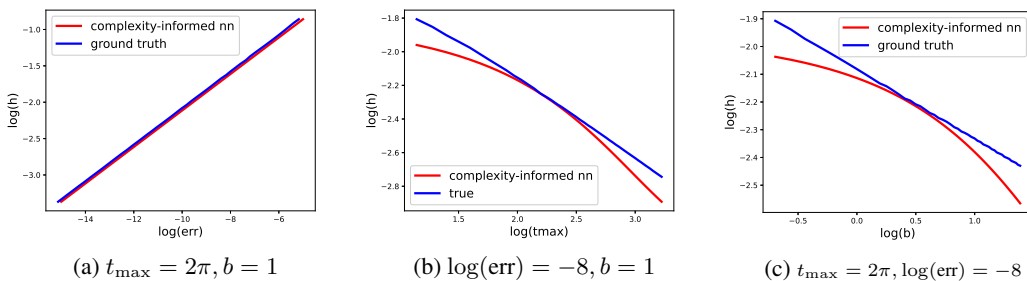

(a) $t_{\max} = 2\pi, b = 1$      (b) $\log(\text{err}) = -8, b = 1$      (c) $t_{\max} = 2\pi, \log(\text{err}) = -8$

Figure 1: Illustration for how $\log h$ changes with respect to different inputs at $x_0 = 0, y_0 = 0$

## 2.2 Time control for adaptive stepsize method

In this subsection, our goal is to provide time guarantee for adaptive stepsize method. We focus on the proportional-integral stepsize controller (PIController) [6], which is a classical control strategy for explicit RK methods. PIController uses the following formula to obtain the new step size $h_{n+1}$

$$h_{n+1} = h_n \left(\frac{\text{sc}}{\text{err}_n}\right)^{\beta_1} \left(\frac{\text{err}_{n-1}}{\text{sc}}\right)^{\beta_2} \tag{7}$$

$\beta_1, \beta_2$ and sc (abbreviation of step control) are three parameters while sc has the greatest influence on the computational time. When $\beta_2 = 0$, the controller is called the integral controller (IController), which only uses $\text{err}_n$, the local error estimation at $t = t_n$, to determine the new stepsize $h_{n+1}$.

To provide time control for the controller in (7), let $Q_1$ be the global error and $Q_2$ be the computational time $T$ in (3), then

$$\begin{aligned}
\min_{\text{sc}} \quad & \text{err}(p, \text{RK}, \text{sc}) \\
\text{s.t.} \quad & T(p, \text{RK}, \text{sc}) \leq T^*
\end{aligned} \tag{8}$$

In (8), $T^*$ is the threshold number of times to evaluate $f(t, u)$ in (1), which is proportional to the computational time approximately.

Similar to the methodology used in subsection 2.1, once $p$ and RK are chosen, an inverse mapping from $T^*$ to sc is required to provide computational time guarantee for adaptive stepsize method. Though adaptive strategy is adopted, by experiments we find the error order is still $O(h^q)$ approximately. Therefore, our goal is to fit the model with the following form

$$\log(\text{sc}) = -k \log T + C(p) \tag{9}$$

After obtaining triples of $(p_i, \text{sc}_i, T_i)$, we can learn the model (9) in training stage and infer the sc given the evaluation times $T$ for adaptive stepsize method.

## 3 Experiments

### 3.1 Fixed stepsize methods

We consider fixed stepsize RK methods with order 2 (Midpoint), 3 (BS3), 4 (RKF4 [7]) and 5 (DP5 [8]). Besides, three different ODE problems are considered: Spiral(6), LotkaVolterra [9] and Brusselator[10]. We generate 6400 data for training, and the size of the test dataset is 1280. The evaluation metric is $R^2$ (coefficient of determination) on test dataset. The quantile loss with quantile value 10% is used to guarantee the error does not surpass the prescribed error level with larger probability. All neural networks used in this experiment are equipped with one hidden layer with size no more than 3. From Table 1 we see that using the complexity-informed model in (5) performs better than using MLP or decision-tree based fitting with the significance level around 5%. Another observation is that incorporating the different RK methods as input to the network model does not decrease the performance of the fitting, as shown in the column of "All" of Table 1. The RK methods are treated as categorical features and pre-processed by one-hot embedding before training.

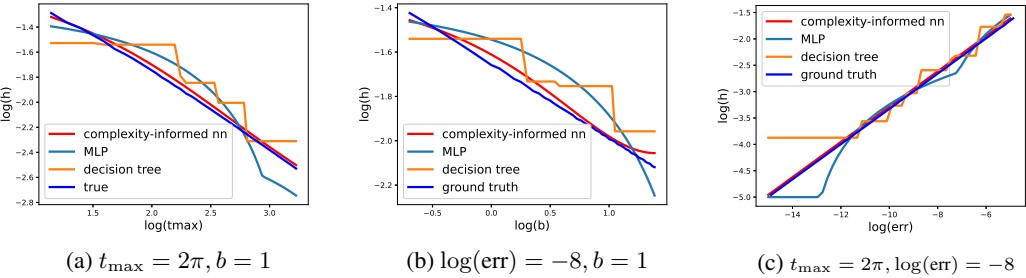

(a) $t_{\max} = 2\pi, b = 1$      (b) $\log(\text{err}) = -8, b = 1$      (c) $t_{\max} = 2\pi, \log(\text{err}) = -8$

Figure 2: Comparison of different fitting methods on Spiral Problem by BS3

The fitted result of different methods are drawn in Figure 2, from which we can see that the curve by the complexity-informed neural network is nearest to that of the ground truth.

Table 1: $R^2$ value for different models

| Problem | Model | RK Methods | | | | |
| --- | --- | --- | --- | --- | --- | --- |
| | | Midpoint | BS3 | RKF4 | DP5 | All |
| Spiral | MLP (baseline) | 0.960 | 0.977 | 0.946 | 0.944 | 0.974 |
| | decision-tree | 0.978 | 0.983 | 0.991 | 0.994 | 0.966 |
| | **complexity-informed** | **0.993** | **0.993** | **0.996** | **0.998** | **0.989** |
| LotkaVolterra | MLP (baseline) | 0.889 | 0.893 | 0.921 | 0.939 | 0.923 |
| | decision-tree | 0.623 | 0.689 | 0.817 | 0.798 | 0.690 |
| | **complexity-informed** | **0.896** | **0.914** | **0.936** | **0.933** | **0.927** |
| Brusselator | MLP (baseline) | 0.930 | 0.952 | 0.966 | 0.892 | 0.933 |
| | decision-tree | 0.920 | 0.893 | 0.906 | 0.839 | 0.865 |
| | **complexity-informed** | **0.972** | **0.972** | **0.964** | **0.929** | **0.962** |

To verify that the predicted stepsize can achieve the desirable error level, we integrate the model into existing ODE solvers and obtain the work-precision diagram for different RK methods with fixed

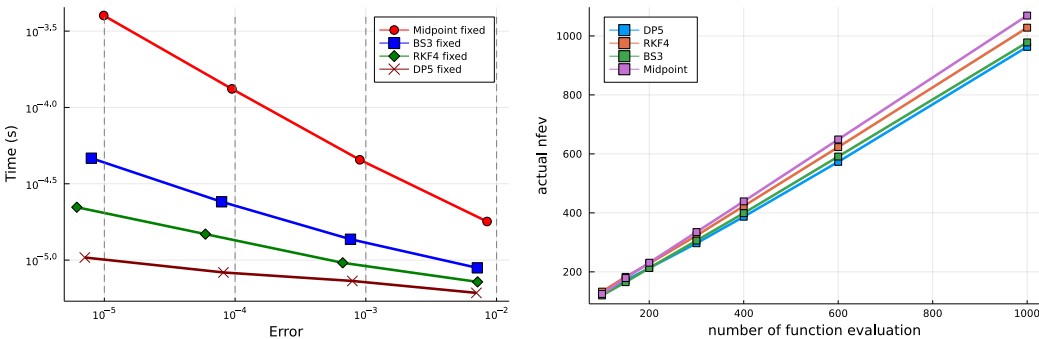

(a) Fixed stepsize RK methods with different orders on the Spiral problem

(b) Time guarantee verification for adaptive stepsize method on the Spiral problem

Figure 3

stepsize strategy. From Figure 3a, we see that for each error level ($10^{-5}$ to $10^{-2}$), the actual error (shown in each mark on the curve) is near the left hand side of the corresponding dotted grey line.

## 3.2 Adaptive stepsize methods

Using the same network architecture as **complexity-informed** in 3.1 and MSE loss, we obtain the fitting results for adaptive stepsize methods with different controllers, which are listed in Table 2.

Table 2: $R^2$ value for different controllers

| Problem | Controller | RK Methods | | | | |
|---|---|---|---|---|---|---|
| | | Midpoint | BS3 | RKF4 | DP5 | All |
| Spiral | IController | 0.995 | 0.999 | 0.988 | 0.953 | 0.996 |
| | PIController | 0.995 | 0.994 | 0.983 | 0.931 | 0.985 |
| LotkaVolterra | IController | 0.968 | 0.932 | 0.873 | 0.729 | 0.926 |
| | PIController | 0.978 | 0.927 | 0.909 | 0.853 | 0.942 |

From Table 2, we see that the model (9) has similar fitting ability for different controllers and fits better for lower order RK methods.

Furthermore, given the expected time $T^*$, we can use the model (9) to get sc and solve the ODE problem to obtain the actual $T$. Then we verify whether the actual evaluation time is equal to the expected time by plotting Figure 3b. We see that all lines are near $T = T^*$ approximately, thus demonstrating the feasibility of our method.

## 4 Future Work

Two directions of further improvement will be considered in the future. Firstly, for now we choose the most decisive variable to guide the fitting of user-oriented target. To further improve the accuracy-efficiency tradeoff, we should include as many solver parameters as possible, such as the order of RK method in (4), and $\beta_1, \beta_2$ in (7). Besides, uniform random sampling is adopted to obtain training data for a specific ODE problem in this paper. In the future, we will explore how to sample more efficiently while keeping the training accuracy, which holds the potential to significantly improve the learning accuracy and efficiency, especially when the ODE systems are of high dimensions.

## Acknowledgment

The work of Shao-Lun Huang was supported in part by the National Natural Science Foundation of China under Grant 61807021, in part by the Shenzhen Science and Technology Program under Grant KQTD20170810150821146.

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
