# OpenReview forum: "Performance-Guaranteed ODE Solvers with Complexity-Informed Neural Networks"
_NeurIPS.cc/2021/Workshop/DLDE — DLDE Workshop -- NeurIPS 2021 Poster_

### Official Review · Reviewer_kvBL · 2021-10-04

**Confidence:** 3

**Review:**

This paper aims to learn technical parameters with time constraints using a learned model mapping from these parameters to the time. The learned model using a complexity-informed method which out-performs MLP or decision-tree based fitting with the significance level around 5%. The approach is interesting, however
- From Figure 1, the fit doesn't look good to me, can the author gives an evaluation metric to see how it's better? Can the author also show the fitted result of the decision tree and MLP in Figure 1?
- For the error in Figure 1 looks the fitted model have error, can the author show that will some of the learned model will exceed the time limit T^*
- More experiment about different selection of T^* can be included in the experiment and different learned hyperparameter of different methods can be listed to understand the benefit of the approach better.
- Have the author tried this method for the learned ODE like Neural ODE?

**Score:**

4: Very good paper

---

### Official Review · Reviewer_F64f · 2021-10-11

**Confidence:** 3

**Review:**

The authors present a novel approach to assess the time vs error trade-off for ODE solvers and tune the solvers parameters accordingly. For this, they present a complexity informed neural network, which is trained to improve the log(err) - log(T) fit. They use this approach to yield optimized parameter choices for a range of experiments. The authors also provide empirical evidence that their method chooses a more optimal step size than two baseline methods.


To my knowledge, this approach to make use of neural networks to tune the parameters of ODE solvers is novel.
The presentation is sound and the experiments show an advantage with respect to other approaches like decision trees and multi-layer perceptrons.
For improved clarity, I would like the authors to include a more thorough explanation of their neural network architectures. Furthermore, the difference between the MLP baseline method and the presented complexity informed method wasn't quite clear to me (apart of the fact that the presented method seems to act in log-space).

**Score:**

4: Very good paper

---

### Official Review · Reviewer_sLw4 · 2021-10-14
**Well written and good technical contribution**

**Confidence:** 4

**Review:**

### Summary
The paper propose a novel approach to learn the optimal step size (or step adaption) of ODE solvers by fitting a neural network to the $\log({\tt err}) -log(h)$ residual. To my knowledge, the learning algorithm presented is novel and potentially impactful if applied to ODE-based machine learning model such as NeuralODEs. The paper is well-written and the experiments showcase the effectiveness of the method.

### Minor Comments

* Can the method be combined to other learning approaches to accelerate inference of ODEs such as `hypersolvers`[1]?

[1] https://arxiv.org/abs/2007.09601

**Score:**

3: Good paper

---

### Decision · Program_Chairs · 2021-10-14

**Decision:**

Accept (Poster)

**Comment:**

Reviews were generally positive, though questions remain about the applicability of the method in systems of higher dimensions. The reviewers suggest discussing the connections between the proposed approach and other learning-to-solve methods, such as hypersolvers.